# Clinical characteristics and outcomes of SARS-Cov-2 B.1.1.529 infections in hospitalized patients and multi-surge comparison in Louisiana

**Katie Taylor**[1,2☯], **Evan Rivere**[1,2☯]*, **Tonya Jagneaux**[1,2‡], **Gabrielle LeBoeuf**[1‡], **Karen Estela**[1‡], **Christi Pierce**[1‡], **Catherine O'Neal**[1,2☯]

1 Franciscan Missionaries of Our Lady Health System, Baton Rouge, LA, United States of America,
2 Louisiana State University Health Sciences Center, Baton Rouge, LA, United States of America

☯ These authors contributed equally to this work.
‡ TJ, GL, KE and CP also contributed equally to this work.
* erive3@lsuhsc.edu

**Data Availability Statement:** All relevant data are within the manuscript.

## Abstract

### Background

Peer reviewed data describing SARS-CoV-2 Omicron variant symptoms and clinical outcomes as compared to prior surges in the United States is thus far limited. We sought to determine disease severity, presenting features, and epidemiologic factors of the SARS-CoV-2 Omicron variant compared to prior surges.

### Methods

Retrospective cohort analysis was performed on patients admitted during five surges in Louisiana between March 2020 and January 2022. Patient data was pulled from the medical record and a subset of patients during Surge 5 were manually abstracted. Patients who were admitted to one of six Louisiana hospitals with a positive SARS-CoV-2 test during the 5 defined surge periods were included. Surges were compared using chi-squared tests and one way ANOVA for age, sex, vaccination status, length of stay, ICU status, ventilation requirement, and disposition at discharge. The records of patients admitted during the omicron surge were analyzed for presenting symptoms and incidental *SARS-CoV-2* diagnosis.

### Results

With each subsequent surge, a smaller proportion of patients presenting to the emergency department were admitted. Patients admitted during surge 5 had shorter lengths of stay and fewer comorbidities than prior surges. Fewer patients in surge 5 presented with a respiratory condition and fewer required ICU admission. In surges 4 and 5, fewer vaccinated patients were admitted compared to their unvaccinated counterparts. Overall mortality was lower in surge 5 (9%) than in surge 4 (15%) p < .0005. Of the SARS-Cov-2 admissions in surge 5, 22.3% were felt to be incidental diagnoses.

**Funding:** The author(s) received no specific funding for this work.

**Competing interests:** The authors have declared that no competing interests exist.

## Conclusions

As the COVID-19 pandemic progressed, a younger and less vaccinated population was associated with higher risk for severe disease, fewer patients required ICU admission and overall mortality decreased. Vaccinations seemed to be protective for overall risk of hospitalization but once admitted did not seem to confer additional protection against severe illness during the omicron surge. Age also contributed to patient outcomes.

## Introduction

The COVID-19 pandemic has been characterized by surges of increased disease activity with intervening periods of reduced activity. The cause of this pattern is complex, as many factors and their interplay influence disease activity: societal and individual behavior, including variability in the use of mitigation; immune status of the population, either vaccinated, previously infected or both; and evolution of the virus, resulting in variants of varying transmissibility and virulence. As a result of these factors and their interactions, each surge has had a unique impact on healthcare systems and outcomes [1, 2].

Recent surges, both globally and nationally, have been characterized by overwhelming spread of the Delta and, most recently, Omicron variants. The Omicron (*B.1.1.529*) variant was first detected in the United States on December 1, 2021 [3, 4], and, by the week ending December 25, 2021, it became the predominant strain nationwide [5, 6]. Cases due to Omicron have been reported to be less severe; however, infections with Omicron in previously recovered and/or fully vaccinated patients have been described, raising concerns of a larger susceptible population [7, 8]. Data from South Africa comparing surges revealed a younger population with fewer admissions and a decreased need for respiratory care during the Omicron-predominant period [9]. In one US hospital's review of Omicron vs Delta surges, the former was associated with lower inpatient mortality; however, no difference was seen in outcome during the Omicron surge with regards to vaccination status [10]. We performed a retrospective cohort analysis of patients admitted to member hospitals of a Louisiana healthcare system during various surges of COVID-19 to examine the differences in volume and outcome of patients admitted with COVID-19. To elucidate the effect of vaccination on outcomes, we compared patients admitted during the Delta and Omicron surges. We also describe presenting symptoms of pediatric (ages 0 to 17) and adult (ages 18 and older) patients admitted to our tertiary referral center during the Omicron predominant surge.

## Materials and methods

The Franciscan Missionaries of Our Lady Health System (FMOLHS) includes six hospitals that serve as regional referral centers in Louisiana and Mississippi; Our Lady of the Lake Baton Rouge, Our Lady of the Lake Ascension, Our Lady of Lourdes, Our Lady of Angels, St. Francis Medical Center and St. Dominic's Hospital. The former five hospitals located throughout Louisiana share a common medical record and were included in the analysis. COVID-19 admissions and emergency department (ED) visits during the time periods corresponding to the individual surges of COVID-19 activity within the state of Louisiana were reviewed and considered; no sample size calculation was performed. Five periods of increased activity were identified as surges: March 18, 2020 to May 1, 2020 (S1); July 3, 2020 to August 24, 2020 (S2); November 28, 2020 to January 30, 2021 (S3); July 10, 2021 to September 25, 2021 (S4, Delta);

and the current surge December 15, 2021 to January 13, 2022 (S5, Omicron). These dates corresponded to the inflection point of increasing and decreasing rates of *SARS-CoV-2* test positivity, except for surge 5 in which data analysis was stopped mid-January to compile results. Widespread sequencing was not performed during the first three surges, however based on public health data* it is assumed that the initial surges were a result of the ancestral COVID-19 variant. The Delta variant became the predominant strain in the United States during the summer of 2021. At the peak of this surge, 98.13% of all Louisiana test isolates were identified as the Delta variant [11]. In the winter of 2021, Omicron became the dominant strain in Louisiana, accounting for 98.99% of sequenced strains at the surge's peak [11].

This retrospective study was approved by the Louisiana State University Health Sciences Center–New Orleans Institutional Review Board (IRB #684) and received a waiver of informed consent for all patients studied. We defined a COVID-19 diagnosis as any PCR or antigen test positive for *SARS-CoV-2* documented within the electronic health record. Patients were only included if the positive test was performed during an ED visit or hospital admission within one of the surge periods. Most tests performed were PCR; however in the early stages of surge 5, PCR testing became limited and antigen testing was used more readily. *SARS-CoV-2* testing was not required for all admissions but was recommended for any patient presenting with symptoms consistent with COVID-19, for patients undergoing an aerosol generating procedure and for patients admitted to mental health locations. Length of stay was calculated by the discharge date, date of death, or by the date when final data collection occurred if the patient was still admitted at that time. The remainder of the data points used for surge-to-surge comparison were obtained through reports generated through the institutions' shared electronic medical record.

Comparison of the surges was performed using excel and chi-squared tests for categorical variables and one-way analysis of variance (ANOVA) for numeric variables [12, 13]. The assumptions of ANOVA including normality, equal variance and independence were met prior to analysis. A p-value less 0.05 was adopted as the level of significance. Variables compared included age, sex, vaccination status, length of stay, ICU status, intubation/ventilation requirement and disposition at discharge. Vaccinations became widely available to the entire U.S. population 18 years or old in March 2021, prior to surge 4, and therefore a separate analysis of vaccination status was performed for surge 4 vs surge 5. For analysis, we classified patients as unvaccinated, overdue for booster, or vaccinated. Unvaccinated patients had no record of vaccination or only received 1 dose of a 2 dose mRNA primary vaccine series. Patients overdue for booster completed a 1 dose virus vector or 2 dose mRNA primary vaccine series and were eligible for an additional immunization but had not yet received the additional dose. Fully vaccinated individuals completed all doses recommended at the time of admission or were not yet due for an additional dose.

A subset of patients admitted between December 15, 2021 and January 7, 2022 to one of FMOLHS's tertiary referral hospitals in Baton Rouge, Our Lady of the Lake Regional Medical Center or Our Lady of the Lake Children's Hospital, was abstracted for admitting symptoms. Symptoms were categorized as respiratory, gastrointestinal, neurologic, and/or cardiac or none. Admit diagnosis was noted. Patients could present with multiple symptoms if documented by the admitting physician. Sepsis was defined as suspected infection and 2 SIRS criteria with at least one criterion being either a qualifying white blood cell count or temperature. Septic shock was defined as sepsis with the need for vasopressor or fluid support to improve hypotension. Patients admitted for surgery, trauma, psychiatric illness, and/or patients whose test-based diagnosis *SARS-CoV-2* infection did not contribute to the reason for hospital admission were categorized as incidental infections.

## Results

### Surge-to-surge comparison

With each subsequent surge, there was a significant decrease in the proportion of COVID-19 patients presenting to the ED who were admitted to the hospital (51%, 43%, 38%, 20%, 15% for S1-S5 respectively, p < .0005) (Table 1). Though admission rates were highest in earlier surges, the later surges (S4 and S5) saw the largest number of *SARS-CoV-2* positive ED visits with S4 having the largest total number of hospital admissions.

Patients admitted during S4 and S5 were younger than those admitted during previous surges (median age [25, 75]: 67[57, 77], 66[52, 76], 69[57, 79], 58[42, 71], and 62[37, 74] for surges 1–5 respectively, p<.0005) (Table 1). Hospital length of stay was lower in S5 compared to previous surges (median [25, 75]: 6[3, 12], 5[2, 10], 5[3, 11], 5[2, 10], 3[2, 17] days for surges 1–5 respectively, p<.0005). The number of admitted patients having comorbidities decreased significantly during S5 compared to the previous surges (62% vs ≥96% S1-S4, p<.0005) and the proportion of patients presenting with a respiratory condition was also significantly lower in S5 vs previous surges (42% vs 57–64% in S1-S4, p < .0005). Significantly fewer patients required an ICU stay in S5 compared to previous surges (25% compared to 34% in S4 and 62% in S1, p<.0005). S4 was associated with the youngest median age at ICU admit and youngest age of death (ICU: 67, 65, 67, 57, 63 for S1-S5 respectively, p < .0005; mortality: 73, 75, 76, 64, 69 for S1-S5 respectively, p < .0005). The number of vaccinated individuals differed between

**Table 1. Comparison of characteristics and outcomes in all patients admitted with COVID-19 over the pandemic.**

| | Surge 1[a] | Surge 2 | Surge 3 | Surge 4 | Surge 5 | p value—all surges | p values—surge 4 vs 5 |
|---|---|---|---|---|---|---|---|
| ED COVID-19 patients, (N) | 1312 | 2462 | 3250 | 7570 | 5233 | | |
| **Admitted,** N (%) | 672 (51) | 1058 (43) | 1227 (38) | 1522 (20) | 787 (15) | < .0005 | < .0005 |
| Sex, M N (%) | 340 (50) | 530 (50) | 617 (50) | 788 (52) | 395 (50) | .599 | .120 |
| Age, median (25, 75) | 67 (57,77) | 66 (52,76) | 69 (57,79) | 58 (42,71) | 62 (37,74) | < .0005 | < .0005 |
| Length of stay (days), median (25, 75) | 6 (3,12) | 5 (2,10) | 5 (3,11) | 5 (2,10) | 3 (2,17) | < .0005 | < .0005 |
| Patients with comorbidities[b], N (%) | 665 (99) | 1029 (97) | 1179 (96) | 1483 (97) | 489 (62) | < .0005 | < .0005 |
| Respiratory condition on admit, N (%) | 381 (57) | 681 (64) | 742 (60) | 948 (62) | 332 (42) | < .0005 | < .0005 |
| ICU admissions, N (%) | 423 (62) | 444 (42) | 454 (37) | 511 (34) | 194 (25) | < .0005 | < .0005 |
| Age, ICU admits, median (25, 75) | 67 (58,76) | 65 (53,74) | 67 (55,77) | 57 (43,69) | 63 (42,72) | < .0005 | .593 |
| Ventilated, N (%) | 181 (27) | 138 (13) | 159 (13) | 214 (14) | 57 (7) | < .0005 | < .0005 |
| **Vaccination status** | | | | | | | |
| Unvaccinated, N (%) | | | | 1287 (85) | 546 (69) | | < .0005 |
| Fully vaccinated[c], N (%) | | | | 235 (15) | 86 (11) | | |
| Overdue for booster[d], N (%) | | | | 0 | 155 (20) | | |
| **Disposition** | | | | | | | |
| Home, N (%) | 273 (40) | 692 (65) | 784 (64) | 1100 (72) | 588 (74) | < .0005 | < .0005 |
| Care facility, N (%) | 199 (30) | 211 (20) | 245 (20) | 198 (13) | 84 (11) | | |
| Expired, N (%) | 200 (30) | 155 (15) | 198 (16) | 224 (15) | 69 (9) | | |
| Not yet discharged, N (%) | | | | | 46 (6) | | |
| Age of Expired, median (25, 75) | 73 (62,82) | 75 (66,84) | 76 (69,82) | 64 (54,75) | 69 (61,80) | < .0005 | .026 |

[a]Surge 1: March 18,2020-May 1, 2020; Surge 2: July 3,2020-August 24, 2020; Surge 3: November 28, 2020-January 30, 2021; Surge 4: July 10, 2021-September 25, 2021; Surge 5: December 15, 2021- January 13, 2022

[b]Comorbidities include diabetes, heart conditions, hypertension, chronic kidney disease, chronic liver disease, chronic pulmonary conditions, and cancer

[c]Fully vaccinated is defined as up to date with primary COVID-19 vaccine series (and booster if recommended) at time of admission.

[d]Overdue for booster is defined as having completed a full 1 or 2 dose primary COVID-19 vaccine series but overdue for additional dose(s) at the time of admission.

S4 and S5 with a significantly higher number of patients admitted who were unvaccinated in S4 compared to S5 (85 vs 69%, p<.0005). Disposition type differed significantly between surges. S1 had the lowest percentage of patients discharged home (40%, p < .0005) and the largest percentage of patients who expired during their hospital stay (30%, p < .0005). There was an overall trend towards decreased mortality surge (30%, 15%, 16%, 15%, 9% for S1-S5 respectively, p < .0005) and decreased need for ventilation (27%, 13%, 13%, 14%, 7% for S1-S5 respectively, p < .0005) as the pandemic progressed (Table 1).

## Adults in the Delta vs Omicron surge comparison by vaccination status

The clinical characteristics and outcomes amongst vaccinated and unvaccinated patients admitted during the Delta (S4) and the Omicron (S5) surges were compared (Table 2). Overall, there were significantly fewer fully vaccinated patients admitted compared to unvaccinated during both surges (17% vs 83% (S4); 11% vs 66% (S5), p = .0005). During S4, unvaccinated patients who were admitted to the hospital, admitted to the ICU, and those who died were significantly younger than vaccinated patients (57 vs 74 (p < .0005); 57 vs 74 (p < .0005); 62 vs 78 (p < .0005), respectively). This age difference was again seen in the S5 surge with the median age of fully vaccinated individuals and of overdue for booster individuals admitted to the hospital being higher than unvaccinated individuals (66 and 68 vs 63, p<.0005). The median age of vaccinated and overdue for booster patients was also higher than that of unvaccinated

**Table 2. Adults ≥ 18 Delta and Omicron surge data by vaccine status.**

| | Surge 4 (Delta variant) | | | Surge 5 (Omicron variant) | | | |
|---|---|---|---|---|---|---|---|
| | Total population | | | Total population | | | |
| | Total patients admitted, 1388 | | | Total patients admitted, 688 | | | |
| Vaccine status, N (%) | Vaccinated[a] 234 (17) | Unvaccinated 1154 (83) | P value <0.005 | Vaccinated[a] 79 (11) | Unvaccinated 454 (66) | Overdue for Booster[b] 155 (23) | P value < .0005 |
| Age, yrs Median (25, 75) | 74 | 57 | < .0005 | 66 | 63 | 68 | < .0005 |
| | (66, 83) | (45, 69) | | (56, 74) | (45, 74) | (62, 78) | |
| Sex, M, N (%) | 123 (53) | 584 (51) | .58 | 41 (51) | 223 (49) | 78 (50) | .95 |
| Patients with comorbidities[c] , N (%) | 225 (96) | 1131 (98) | .085 | 54 (68) | 295 (65) | 88 (57) | .119 |
| Length of stay Median (25,75) | 4 | 5 | .004 | 4(2,5) | 4 | 5 | .06 |
| | (3, 9) | (3, 10) | | | (2,8) | (2,9) | |
| **ICU status**, N (%) | 66 (28) | 404 (35) | .04 | 16 (20) | 108 (28) | 43 (28) | .41 |
| Age, ICU admits Median (25, 75) | 74 | 57 | < .0005 | 65 | 63 | 66 | .03 |
| | (70, 86) | (44, 68) | | (57,72) | (59,73) | (59, 77) | |
| Ventilated, N (%) | 17 (7) | 185 (16) | < .0005 | 5 (6) | 33 (7) | 15 (10) | .88 |
| **Disposition** | | | | | | | |
| Home, N (%) | 141 (60) | 831 (72) | < .0005 | 64 (81) | 327 (72) | 105 (68) | .50 |
| Care Facility, N (%) | 55 (24) | 140 (12) | | 8 (10) | 53 (12) | 21 (14) | |
| Expired, N (%) | 38 (16) | 183 (16) | | 4 (5) | 46 (10) | 19 (12) | |
| Not yet discharged, N (%) | | | | 3 (3) | 28 (6) | 10 (6) | |
| Age of Expired Median (25, 75)) | 78 | 62 | < .0005 | 73 | 67 | 71 | .29 |
| | (71, 86) | (52, 72) | | (70, 76) | (58, 80) | (66, 82) | |

[a]Vaccinated is defined as up to date with primary COVID-19 vaccine series (and booster if recommended) at time of admission.

[b]Overdue for booster is defined as having completed a full 1 dose virus vector or 2 dose mRNA primary COVID-19 vaccine series but overdue for additional dose(s) at the time of admission.

[c]Comorbidities include diabetes, heart conditions, hypertension, chronic kidney disease, chronic liver disease, chronic pulmonary conditions, and cancer

patients admitted to the ICU during the S5 surge (65, 66, 63 respectively, p = .03). There was no difference in age by vaccination status for inpatient mortality during S5 (p = .29), although the overall mortality was lower in S5 (15% vs. 9%, p < .0005)(Table 1). Length of stay was significantly different between vaccinated and unvaccinated individuals in S4 (4 vs 5 days, p = .004) but did not reach statistical significance in S5 (4 days fully vaccinated, 4 days unvaccinated vs 5 days overdue for booster, p = .06).

A higher percentage of patients required ventilation during S4 compared to S5 (14% vs 7%, p < .0005). Though vaccinated patients required less ventilatory support in S4 surge (7% vaccinated vs 16% unvaccinated, p<.0005), there was no difference in ventilatory support by vaccination status in S5. The percent of patients with comorbidities did not differ by vaccination status in either surge.

## Omicron patient characteristics

In *SARS-CoV-2* positive patients during the Omicron surge (S5), respiratory complaints were present in 64% of adults, with gastrointestinal and neurologic symptoms occurring in greater than 20% of patients. 25.8% of adult patients presented with sepsis and 10.4% of adult patients presented with septic shock (Table 3).

In the S5 *SARS-CoV-2* positive pediatric population, 21.5% of children required an ICU admission with 3% of children requiring mechanical ventilation. The most common presenting symptom among pediatric patients was respiratory (52%) followed by gastrointestinal

**Table 3. Characteristics of pediatric and adult patients admitted with Sars-CoV-2 during the Winter 2021–22 surge.**

| N (%) | | |
|---|---|---|
| **Patients** | **Total population ≤17 N = 65** | **Total population ≥18 N = 337** |
| Age, yrs (median) | 1 | 63 |
| Sex, M | 34 (52) | 170 (50) |
| Length of stay | 2 | 3.1 |
| ICU admissions | 14 (21.5) | 63 (18.6) |
| Fully Vaccinated | 0 | 46 (14) |
| Ventilated | 2 (3) | 16 (5) |
| Mortality | 0 | 9 (2.7) |
| **Presenting signs/symptoms** | | |
| Respiratory | 34 (52) | 215 (64) |
| Gastrointestinal | 23 (35) | 76 (22.5) |
| Cardiac | 0 | 42 (12.5) |
| Neurologic | 4 (6) | 75 (22.3) |
| Sepsis | 24 (37) | 87 (25.8) |
| Septic Shock | 3 (5) | 35 (10.4) |
| Coagulopathy | 0 | 23 (7) |
| Incidental covid | 11 (7) | 79 (23.4) |

Definition of signs/symptoms: respiratory–cough, stridor, shortness of breath, respiratory distress, infiltrates on chest film, hypoxia or hypoxemia; cardiac–chest pain, arrythmia, myocarditis, pericarditis, myocardial infarction, heart failure exacerbation; gastrointestinal–nausea, vomiting, diarrhea, appendicitis, gastrointestinal bleed; coagulopathy–arterial or venous thrombosis involving any organ system; sepsis– 2 SIRS criteria met with at least 1 criterion being either white blood cell count or temperature; septic shock–sepsis with hypotension requiring vasopressor or fluid support

(35%). Only 7% of pediatric admits were found to be incidental diagnoses compared to 23% of adult admissions (Table 3).

Within the group of *SARS-CoV-2* positive patients, 5 (1%) patients were diagnosed with appendicitis of which 4 were between the ages of 11 and 16, and 11 (3%) patients presented with seizure, 2 of which were under the age of 1, and 8 of which were new in onset. Additionally, we noted 19 (5%) presenting with atrial fibrillation with rapid ventricular rate, 10 (2%) presenting with sickle cell vaso-occlusive crisis and 7 (2%) with gastrointestinal bleeding.

## Discussion

The Omicron surge began in the United States in December 2021. Following this variant's introduction, hospitals and emergency departments became overwhelmed with patients as daily death rate and hospitalization rates climbed, similar to previous COVID-19 surges. Although the Omicron surge resulted in the lowest admission rates of all surges (15%), this was counterbalanced by it yielding the second-highest number of *SARS-CoV-2* positive ED visits. In all, S5 resulted in more inpatient admissions than S1, despite a truncated analysis due to data collection.

The surge-to-surge comparison revealed an overall younger population and the less vaccinated population at risk for severe disease as the pandemic progressed coinciding with data out of South Africa and California [9, 10]. ICU admissions, percent of patients requiring ventilation, and percent of patients who expired while admitted all decreased with subsequent surges. These findings coincide with the findings from South Africa, which showed smaller percentage of patients requiring mechanical ventilation and admission to ICU during the omicron wave as compared to the delta wave [9]. Several factors likely contribute to this finding. Importantly, the age of admission, a powerful predictor of outcomes, generally declined with successive surges. Also, as clinicians gained experience in caring for COVID-19 patients and therapy choices improved, evolving medical care likely influenced outcomes. As the proportion of patients who were vaccinated or immune by prior infection increased–and as this acquired immunity waned over time–the host response played an important role in influencing outcomes. In addition, we were not able to capture the number of patients who were diagnosed and treated at home once home testing and improved ambulatory therapy was available. All these factors likely affected inpatient outcomes. In short, the relationship between each of these factors and outcome is more difficult to elucidate.

Changes in hospital practice also influenced certain clinical outcomes. For example, due to resource limitation during S4, the hospital policy allowed for up to 30 liters per minute of high-flow oxygen delivered by heated, humidified nasal cannula to be cared for on the general inpatient wards, thus underestimating the number of critically ill patients. In effect, to a degree, this altered standard of care during S4 uncoupled the relationship between severity of illness and location of admission. This was not a common practice during previous or subsequent surges, and it is unclear if this practice may have influenced other clinical outcomes (mortality, length of stay, etc). The percent of patients admitted with primary respiratory condition per ICD10 code, decreased as the pandemic progressed, and this decrease was most pronounced in S5 compared to previous surges. This may have been due to an evolution in presentation of COVID-19 in the vaccinated population however the number of incidental COVID-19 diagnoses made during S5 (>20%) may also have played a role. There were significantly more patients admitted during S5 who had no comorbidities, potentially indicating that our incidental number of COVID-19 diagnoses also increased over previous. Unfortunately, our individual chart review for incidental numbers only included S5 and we cannot fully compare incidental diagnoses from previous surges.

In comparing the five surges within our health system, two overarching observations were noted: vaccination is protective against severe disease and age plays a significant role in outcome severity. Vaccination was clearly protective for overall risk of hospitalization in the Delta and Omicron surge; however, vaccination did not appear to be associated with the same degree of protection against severe illness once admitted (defined as ICU admission, mechanical ventilation, disposition, or death) during S5 compared to S4 (Table 2). The overall outcomes of patients in S5 were improved versus S4 with fewer patients discharged to a care facility and lower mortality. 31% of patients admitted during the Omicron surge had received a primary series or a series plus booster compared to 15% of patients during the Delta surge. Although we are unable to determine if variant type played a part in outcome, our surge-to-surge comparison reveals that overall vaccination rate significantly increased during the Omicron surge and likely played a role in overall outcomes. Additional factors that changed as the pandemic progressed such as improved medical knowledge, standardized practices, community awareness as well as advancing therapeutics likely also played a role in outcomes.

Age continued to be a defining factor for outcome in S5 as it was in S4. The age of those admitted with COVID-19 decreased as the pandemic progressed and may have been protective for some outcomes. There was a protective effect of younger age in disposition between vaccination groups in S4. More unvaccinated patients were able to be discharged home during S4; however, their median age was 17 years younger than their vaccinated counterparts. Despite the large age gap, mortality was similar between the younger unvaccinated patients and the more elderly, vaccinated population during S4 supporting the protective effect of the vaccine. During S5, vaccination appeared to trend towards more protection than age as the vaccinated group was only 3 years older than the unvaccinated group, and vaccinated patients were more likely to be discharged to home. However, this difference did not reach statistical significance. Additionally, during S5, the vaccinated patients requiring ICU admission were only 2 years older than vaccinated patient and were 8% less likely to require ICU admission, albeit this percentage difference did not reach statistical significance. And while also not statistically different, the vaccinated group was only 6 years older but 5% less likely to die than the unvaccinated group during S5.

Previous pediatric data collected from five pediatric hospitals during the Delta surge showed 29.5% of pediatric patients with COVID-19 required an ICU admission (14) [14]. In the same study, the authors reported a 19% incidental diagnosis rate. In our review of pediatric Omicron data, we also saw a >20% rate of ICU admissions in children with respiratory symptoms, gastrointestinal symptoms, and sepsis being the most common admission diagnoses. However, only 7% of pediatric cases were incidental suggesting that our admitted pediatric patient population during Omicron was more likely admitted with symptomatic illness.

Patients with acute symptomatic COVID-19 present most commonly with respiratory symptoms, but often multiple organ systems are involved in the disease and patients can occasionally present solely with non-respiratory symptoms [15–17]. In our adult population, gastrointestinal symptoms and sepsis were common presenting findings during S5. Previous literature supports links between COVID-19 and appendicitis, seizures, gastrointestinal bleeding, atrial fibrillation, and sickle cell vaso-occlusive crisis [18–26], and these associations were supported by our findings [18–26].

This study has several limitations. We did not have individual variant analysis for each patient and therefore cannot determine if patient outcomes are directly related to variant affect. We did not perform a multivariate analysis which may help us to determine the weight of individual factors associated with outcomes. A multivariate analysis would be difficult in this size study as age, vaccination and comorbidities may trend together in population subsets. Further studies would need to be performed to elucidate the effect of variant vs. vaccination

status in patient outcomes. Selection bias may be present given that COVID-19 testing was not performed on all admitted patients. Testing modality was not uniform throughout the entire pandemic therefore detection rate may have been affected by various test types. Additionally, patients diagnosed and treated at home were not included in this analysis. Omicron data was collected near the peak of the surge and not through the end of the surge. Total admitted patients, length of stay and mortality are likely underrepresented when compared to surge 1 through surge 4, where data collection included the entire time interval of each surge. Thus, the numbers may be underpowered to find a significant difference.

Despite the limitations, this study reveals the challenges hospitals faced in anticipating patient care as each surge occurred. The differences between prognosis of patients in the Delta vs Omicron surge were significant. However, with 25% of the Omicron admissions requiring ICU care, an older age of admission compared to Delta and 9% inpatient mortality at the time of our analysis, the severity of illness and anticipated effect of the Omicron variant on hospital capacity was initially underestimated. Our study reveals not only the immense resources that each surge has required but also the variety of patient presentations and diverse organ system specific care required for COVID-19 patients.

## Conclusion

Subsequent surges with potentially new variants are an expected reality, leaving us with two recommendations: during increased community activity of *SARS-CoV-2*, acute care evaluation should include COVID-19 testing, given the variability of organ system involvement at presentation. In addition, despite differences in variant characteristics, hospitals should brace for high admission rates and mortality with subsequent surges.

## Acknowledgments

The authors thank Dr. Hollis O'Neal.

## Author Contributions

**Conceptualization:** Katie Taylor, Evan Rivere, Tonya Jagneaux, Karen Estela, Christi Pierce, Catherine O'Neal.

**Data curation:** Katie Taylor, Tonya Jagneaux, Karen Estela, Catherine O'Neal.

**Formal analysis:** Katie Taylor, Evan Rivere, Gabrielle LeBoeuf, Catherine O'Neal.

**Investigation:** Katie Taylor, Evan Rivere, Karen Estela, Christi Pierce, Catherine O'Neal.

**Methodology:** Katie Taylor, Evan Rivere, Tonya Jagneaux, Gabrielle LeBoeuf, Catherine O'Neal.

**Project administration:** Katie Taylor.

**Resources:** Tonya Jagneaux, Christi Pierce.

**Software:** Tonya Jagneaux, Gabrielle LeBoeuf, Karen Estela.

**Supervision:** Christi Pierce.

**Validation:** Katie Taylor, Catherine O'Neal.

**Writing – original draft:** Katie Taylor, Evan Rivere, Catherine O'Neal.

**Writing – review & editing:** Katie Taylor, Evan Rivere, Catherine O'Neal.

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
