## [Decision Letter · Decision Letter 0]

20 Jun 2022

PONE-D-22-13503Multi-surge comparison of COVID-19 characteristics and outcomes of hospitalized patients in LouisianaPLOS ONE

Dear Dr. Rivere,

Thank you for submitting your manuscript to PLOS ONE. After careful consideration, we feel that it has merit but does not fully meet PLOS ONE’s publication criteria as it currently stands. Therefore, we invite you to submit a revised version of the manuscript that addresses the points raised during the review process.

We look forward to receiving your revised manuscript.

Kind regards,

Dong Keon Yon, MD, FACAAI

Academic Editor

PLOS ONE

Journal Requirements:

3. Please amend the manuscript submission data (via Edit Submission) to include author Katie Taylor,Tonya Jagneaux, Gabrielle LeBoeuf, Karen Estela, Christi Pierce and Catherine O’Neal.

Additional Editor Comments:

Thank you for submitting your manuscript to Plos One. The reviewers and I believe it is of potential value for our readers. However, the reviewers have raised a number of very important issues, and their excellent comments will need to be adequately addressed in a revision before the acceptability of your manuscript for publication in the Journal can be determined.

Reviewers' comments:

Reviewer's Responses to Questions

**Comments to the Author**

1. Is the manuscript technically sound, and do the data support the conclusions?

Reviewer #1: Yes

Reviewer #2: Partly

Reviewer #3: Partly

2. Has the statistical analysis been performed appropriately and rigorously? 

Reviewer #1: Yes

Reviewer #2: I Don't Know

Reviewer #3: Yes

3. Have the authors made all data underlying the findings in their manuscript fully available?

Reviewer #1: Yes

Reviewer #2: Yes

Reviewer #3: Yes

4. Is the manuscript presented in an intelligible fashion and written in standard English?

Reviewer #1: Yes

Reviewer #2: Yes

Reviewer #3: Yes

5. Review Comments to the Author

Reviewer #1: Dear Author (s);

The manuscript attempts to determine disease severity, presenting features, and epidemiologic factors of the SARS-CoV-2 Omicron variant compared to prior surges in four hospitals located in Louisiana; the authors found many interested and significant associations. Although the use of retrospective and considerably large dataset over a long period of time (March 18, 2020- January 13, 2022) is little bit questionable, the manuscript builds on a good body of research on this topic. The analyses are appropriate and the conclusions are mostly well supported by the results. It is also quite well written, apart from lack of multivariate analysis. The authors make clear the objectives of the research.

Regards

Reviewer #2: This is an interesting paper aimed to describe differences in characteristics of hospitalized patients among the different surges of COVID-19 in Louisiana, USA. Although it is a selection of hospitals, findings are relevant gave the need of improve knowledge about COVID-19. Despite that, some issues are needed to be clarified and improved to consider the publication at Plos One.

Major comments

I would like to clarify how the authors differentiate a patient unvaccinated partially vaccinated from one overdue for booster? Should not the former been eligible for a booster? (line 118)

How was defined the subset of patients who have symptoms? Is it a sample (and how was it defined) or they are all the patients with symptoms during the dates described? (Line 123)

It is needed to clarify if the assumptions to use a parametric test (ANOVA) were accomplished.

Discussion: Although it is an interesting discussion, most of the statements are done apparently in based the experience in the hospital, missing the comparison of the findings with other experiences (and their corresponding citations). A discussion about why children could be being more admitted during surge 5 could be included. Some of the results have been developed in the discussion section more than discussed as such in that section.

I was wondering how the schemes of vaccination in USA are compatible with the timeline of the surges and the definitions of complete or not scheme, in order to interpret the findings (for example, line 174) and potential effects in the population of an important percentage of vaccinated population.

Specific comments

Median in tables should be expressed as RIC or percentile (25-75, for example) (not Q1-Q3). N(%) should be in the row since there are medians in some rows.

Line 157: Interval of length of stay in surge 5 differs between the text and the table.

Line 158 should be ≥96%.

Line 186: is correct the use of the term clinical significance in that phrase? I think the authors are talking about statistical significance.

Line 195: is it 14 and 7%, or 15 and 8%?

Line 235: Vaccination status is not comparable surge to surge (at least for the first surges).

Minor comments

Some references should be reviewed. For example, link to reference 8 does not seem to be correct.

Reviewer #3: I would like to express sincere gratitude for the chance to review the manuscript.

Summary

This study describes the omicron variant surge impacts on a health system and compares its characteristics to previous surges. The authors demonstrated a younger, less vaccinated population had a higher risk of serious illness, ICU hospitalization rate, and overall mortality have decreased during the Omicron surge (S5) compared to the previous surges. I read it with great interest, but there are some points to be considered before publishing this manuscript.

Major comments

● During the omicron surge, the vaccination rate increased. I think it is hard to determine that the omicron variant decreased the risk of serious illness because of the omicron variant characteristics themselves. How did you control the vaccination effect? This article would be hard to guarantee acceptance unless the authors show additional data analysis of the vaccination effect itself on the omicron.

● Line 103: Can you be more specific? What kind of test for “any test”? (PCR, rapid antigen test?) Did you use the same Covid test for all samples? If not, different methods might affect the result because the detection rate of the test and accuracy are different.

Minor comments

● Why the title does not represent the omicron surge even though the background of the study only talks about the omicron variants?

Abstract

● Lines 28,36: write out a term of FMOLHS, ED (emergency department)

● Line 38: show quantitative data with p-values and the number of samples at each surge

● Line 29: discuss how each surge is defined. what criteria did you follow

● Line 28,35,38: match the terminology for each surge (surge 5, s5). I suggest using surge 5 instead of s5

● Line 41: I don’t understand why you suddenly mention the “younger” population. If you want to add the age variable, this also has to be explained in the method.

Introduction

● Overall, there is a lack of reference in the first paragraph. Make sure to refer to a thesis that can support the statement.

● The hypothesis is missing at the end of the introduction.

● Line 80: Provide the age range for pediatric patients

Method & Material

● Line 84: List five hospitals that were involved.

● Lines 89-92: How did you define the dates for each surge? need the references for each period

● Line 91: Be consistent with how you write the dates.

● Lines 96-97: Add references. Be specific with where the data is obtained. Is it from the Lousiana Department of Health as indicated by the asterisk? If so, why is it “assumed” if it is unbiased data obtained from the public institution?

● Line 102: Include the IRB protocol number to show the study has been approved by IRB

● Lines 106-107: “SARS106 CoV-2 testing was not required for all admissions but was recommended for any patient 107 presenting with symptoms consistent with COVID-19” -> Since I assume there is a selection bias in this study, this needs to be added to the conclusion as a limitation (Lines 300-308)

● Line 119: needs to explain further what kind of vaccine the patients got. (i.e., mRNA, virus vector) Because the Jassen vaccine only requires one shot as opposed to others

● Line 127: Revise the definition. There should be an updated version.

● Lines 128-129: How did you define the septic shock? I guess using EMR data would be hard to code. Using EMR data can lead the sepsis data to be considered a septic shock because of the fluid support. Didn’t you use the ICD code for data analysis? If so, the definition of the septic shock should be changed

● Explain how you calculate the sample size. A flow diagram with numbers of invited, enrolled, and excluded subjects would be helpful to the readers.

● Add the phrase for statistical significance of p-value (i.e., a p-value lower than 0.05 was adopted as the level of significance).

● The authors have to cite the paper of statistical method guideline (i.e., DOI: https://doi.org/10.54724/lc.2022.e3)

Results

● Lines 153,156,185: overused the word “significantly”

● Lines 175-185: Use a consistent format of indicating p-values

(line 175 p. 0.005 line 177 p <.0005 p<.0005 line 182 p .03 line 183 p. 0.29)

● Table 2: The left side is left-aligned, but the right side of the table is center-aligned. It would be better to match the text alignment

● Table 2, Line 146: Use CCI (Charlson comorbidity index) for the column ‘patients with comorbidities vaccination status’ & describe it in the method part. Also, describe the vaccination type (mRNA, virus)

Discussion

● Line 228: add a comma after the introduction

● Line 230: “the” lowest

● Lines 243 - 246: Not including the patients diagnosed and treated at home needs to be commented in the limitation

● Line 248: write out what LPM stands for before using it

● Line 254: identify what “primary respiratory symptoms” are

● Line 266: Add the reference to support with quantitative evidence

● Line 269: Change Thirty-one percent to 31%

● Lines 271-3: this sentence contradicts your argument above in regards to COVID variation. What is the purpose of this sentence?

● Line 273: vaccination rate not vaccination status; increased not improved

● Line 281: except mortality during S5? line 279?

● Line 280,281: Be consistent. Use either S5 or Omicron. Mixed terminology may confuse the reader.

● Line 287: Change parentheses to bracket and put period after the bracket

Conclusion

● Line 320: Observation? It’s rather a recommendation/suggestion

6. PLOS authors have the option to publish the peer review history of their article (what does this mean?). If published, this will include your full peer review and any attached files.

Reviewer #1: No

Reviewer #2: No

Reviewer #3: No

---

## [Author Response · Author response to Decision Letter 0]

7 Aug 2022

Responses to Reviewer #2 Comments:

1. I would like to clarify how the authors differentiate a patient unvaccinated partially vaccinated from one overdue for booster? Should not the former been eligible for a booster? (line 118) 

a. Author comment - We wanted to differentiate a patient who had not completed a recommended primary series from one who had but had not received a booster. Those who had not completed a primary series were included in the unvaccinated category for our analysis. We have clarified the sentence 118 to help to define the categories better. 

b. Revision (line 132) - Unvaccinated patients had no record of vaccination or only received 1 dose of a 2 dose mRNA primary vaccine series. Patients overdue for booster completed a 1 dose virus vector or 2 dose mRNA primary vaccine series and were eligible for an additional immunization but had not yet received the additional dose.

2. How was defined the subset of patients who have symptoms? Is it a sample (and how was it defined) or they are all the patients with symptoms during the dates described? (Line 123)

a. Author comment - Our hospital system has several hospitals. Two of those our tertiary referral centers in Baton Rouge, namely Our Lady of the Lake Regional Medical Center or Our Lady of the Lake Children’s Hospital. The subset is all patients admitted with COVID-19 to either of these hospitals, regardless of symptoms, during the defined time period. We have adjusted the description below for clarity. 

b. Revision (line 138) - A subset of patients admitted between December 15, 2021 and January 7, 2022 to one of FMOLHS’s tertiary referral hospitals in Baton Rouge, Our Lady of the Lake Regional Medical Center or Our Lady of the Lake Children’s Hospital, was abstracted for admitting symptoms. Symptoms were categorized as respiratory, gastrointestinal, neurologic, and/or cardiac or none.

3. It is needed to clarify if the assumptions to use a parametric test (ANOVA) were accomplished.

a. Revision (line 125) - The assumptions of ANOVA including normality, equal variance and independence were met prior to analysis.

4. Although it is an interesting discussion, most of the statements are done apparently in based the experience in the hospital, missing the comparison of the findings with other experiences (and their corresponding citations). A discussion about why children could be being more admitted during surge 5 could be included. Some of the results have been developed in the discussion section more than discussed as such in that section.

a. Author comment: We did not include surge to surge comparison of pediatric cases (ED, admission, etc) and therefore we did not comment on the number of omicron pediatric admissions. Our purpose was to characterize those pediatric admissions we did see during omicron, not quantify. However, we did revise to include reference to other studies which characterized omicron based on prior surge comparisons. 

b. Revisions (lines 257-263) - The surge-to-surge comparison revealed an overall younger population and the less vaccinated population at risk for severe disease as the pandemic progressed coinciding with data out of South Africa and California [9,10]. ICU admissions, percent of patients requiring ventilation, and percent of patients who expired while admitted all decreased with subsequent surges. These findings coincide with the findings from South Africa, which showed smaller percentage of patients requiring mechanical ventilation and admission to ICU during the omicron wave as compared to the delta wave [9].

5. I was wondering how the schemes of vaccination in USA are compatible with the timeline of the surges and the definitions of complete or not scheme, in order to interpret the findings (for example, line 174) and potential effects in the population of an important percentage of vaccinated population.

a. Revision/addition to methods (line 128): Vaccinations became widely available to the entire U.S. population 18 years or old in March 2021 prior to surge 4 and therefore a separate analysis of vaccination status was performed for surge 4 vs surge 5. (Methods)

6. Median in tables should be expressed as RIC or percentile (25-75, for example) (not Q1-Q3). N(%) should be in the row since there are medians in some rows.

a. Revision (Tables 1 and 2) – Median Q1, Q3 changed to percentiles = and N(%) has been properly allocated to the appropriate rows

7. Line 157: Interval of length of stay in surge 5 differs between the text and the table.

a. Revision (line 173) - Hospital length of stay was lower in S5 compared to previous surges (median [25, 75]: 6[3, 12], 5[2, 10], 5[3, 11], 5[2, 10], 3[2, 17] days for surges 1-5 respectively, p< .0005).

8. Line 158 should be ≥96%.

a. Revision (line 175) - The number of admitted patients having comorbidities decreased significantly during S5 compared to the previous surges (62% vs ≥96% S1-S4, p<.0005) and the proportion of patients presenting with a respiratory condition was also significantly lower in S5 vs previous surges (42% vs 57-64% in S1-S4, p<.0005).

9. Line 186: is correct the use of the term clinical significance in that phrase? I think the authors are talking about statistical significance.

a. Revision (line 204) - Length of stay was significantly different between vaccinated and unvaccinated individuals in S4 (4 vs 5 days, p=.004) but did not reach statistical significance in S5 (4 days fully vaccinated, 4 days unvaccinated vs 5 days overdue for booster, p=.06).

10. Line 195: is it 14 and 7%, or 15 and 8%?

a. Author comment: We feel that these percentages are correct based on calculations below. See table 1, rows “expired” and “admitted” for surge 4 and surge 5 columns for N values. Percentage = Total expired / total admitted * 100% (rounded to nearest whole percentile). 

i. Surge 4 mortality: 224/1522 = 0.147 = 15%

ii. Surge 5 mortality: 69/787 = 0.087 = 9%

b. Author comment: 14 vs. 7% is the percentage of patients requiring ventilation (see line 218)

11. Line 235: Vaccination status is not comparable surge to surge (at least for the first surges). Katie and please pair with other comment

a. Author response: Vaccination was not widely available in the United States until January of 2021 therefore was not widely available during the first 3 surges, however the comments made reflect the age differences and mortality/ventilation/ICU admits seen before (surges 1-3) and after (surges 4-5) vaccination became available. See lines 170, 178, and 186 for data to support this observation. 

b. Line 170 - Patients admitted during S4 and S5 were younger than those admitted during previous surges (median age [25, 75]: 67[57, 77], 66[52, 76], 69[57, 79], 58[42, 71], and 62[37, 74] for surges 1-5 respectively, p<.0005) (Table 1).

a. Line 178 - Significantly fewer patients required an ICU stay in S5 compared to previous surges (25% compared to 34% in S4 and 62% in S1, p<.0005).

b. Added Line 186 - There was an overall trend towards decreased mortality surge (30%, 15%, 16%, 15%, 9% for S1-S5 respectively) and ventilation (27%, 13%, 13%, 14%, 7% for S1-S5 respectively) as the pandemic progressed.

c. The surge-to-surge comparison revealed an overall younger population and the less vaccinated population at risk for severe disease as the pandemic progressed.

12. Some references should be reviewed. For example, link to reference 8 does not seem to be correct.

a. Revision – Two incorrect hyperlinks found and corrected. Entire bibliography updated from Mendeley.

Responses to Reviewer #3:

1. During the omicron surge, the vaccination rate increased. I think it is hard to determine that the omicron variant decreased the risk of serious illness because of the omicron variant characteristics themselves. How did you control the vaccination effect? This article would be hard to guarantee acceptance unless the authors show additional data analysis of the vaccination effect itself on the omicron. 

a. Author comment - We acknowledge that our analysis and any analysis will be difficult to determine vaccine effect vs variant characteristic because baseline immune status in the population admittedly is unknown. There is a clarifying statement in line 298 to let the reader know that we are not attempting to define the variant characteristics in this paper. 

b. Line 298 - The variant type may have played a part in outcome, but the surge-to-surge comparison also reveals that overall vaccination status is significantly improved during the Omicron surge and may also have played a role in overall outcomes. 

c. We have also added this as a limitation in our conclusion (line 339) - Further studies would need to be performed to elucidate the effect of variant vs. vaccination status in patient outcomes.

2. Line 103: Can you be more specific? What kind of test for “any test”? (PCR, rapid antigen test?) Did you use the same Covid test for all samples? If not, different methods might affect the result because the detection rate of the test and accuracy are different. 

a. Author common - Different tests were accepted including PCR, multiplex PCR, rapid antigen. Antibody testing was not used. The majority of hospital based testing was PCR based testing during surge 1-5 however antigen testing was intermittently used in the later part of surge 5 due to limited PCR tests at that time. If a clinical discrepancy was noted in testing, repeat testing was performed by PCR as per hospital policy.

b. Revision (line 111) - We defined a COVID-19 diagnosis as any PCR or antigen test positive for SARS-CoV-2 documented within the electronic health record. 

c. Revision/addition (line 114) - Most tests performed were PCR; however, in the early stages of surge 5, PCR testing became limited therefore antigen testing was used more readily

3. Why the title does not represent the omicron surge even though the background of the study only talks about the omicron variants?

a. Author comment: There were two versions of the title initially proposed. The title was chosen as it was more concise, however it omitted our focus on the omicron variant. We have adjusted the title to the alternative version to reflect the focus on the omicron variant.

b. Revision (Line 1) - Clinical Characteristics and Outcomes of SARS-Cov-2 B.1.1.529 Infections in Hospitalized Patients and Multi-Surge Comparison in Louisiana

4. Lines 28,36: write out a term of FMOLHS, ED (emergency department) 

a. Revisions (line 30) - Patients who were admitted to a Baton Rouge hospital with a positive SARS-CoV-2 test

b. Revision (line 37) - a smaller proportion of patients presenting to the emergency department

5. Line 38: show quantitative data with p-values and the number of samples at each surge

a. Author comment – Because of the limited word count of 300 in the abstract, we did not included the N with the percentages but did however edit to include the p value. The data is readily available table 1 and results section of the manuscript.

b. Revision (line 38) - Overall mortality was lower in surge 5 (9%) than in surge 4 (15%) p<.0005

6. Line 29: discuss how each surge is defined. what criteria did you follow

a. Author comment – See line 99 of the methods section. Because of the limited word count in the abstract, we did not expound on this until the methods section of the manuscript. 

7. Line 28,35,38: match the terminology for each surge (surge 5, s5). I suggest using surge 5 instead of s5

a. Revisions made to use surge instead of s

8. Line 41: I don’t understand why you suddenly mention the “younger” population. If you want to add the age variable, this also has to be explained in the method.

a. Line 44 - As the COVID-19 pandemic progressed, a younger and less vaccinated population was at higher risk for severe disease, fewer patients required ICU admission and overall mortality decreased.

b. Age is included in the methods section of abstract: (Line 32) Surges were compared using chi-squared tests and one way ANOVA for age, sex, vaccination status, length of stay, ICU status, ventilation requirement, and disposition at discharge.

c. Age is also included in the methods section of the manuscript (line 126) - Variables compared included age, sex, vaccination status, length of stay, ICU status, intubation/ventilation requirement and disposition at discharge.

9. Introduction: Overall, there is a lack of reference in the first paragraph. Make sure to refer to a thesis that can support the statement.

a. Author comment – Two studies added for reference to show that surges have been shown to exist in other studies and that their impact on healthcare is unique for each surge.

b. Revision (line 66): As a result of these factors and their interactions, each surge has had a unique impact on healthcare systems and outcomes [1,2]. 

c. Seasonal COVID-19 surge related hospital volumes and case fatality rates | BMC Infectious Diseases | Full Text (biomedcentral.com) 

d. A Tale of Two Surges: Differences in Outcomes in the COVID-19 Pandemic in a Community Teaching Hospital in Massachusetts - PMC (nih.gov) 

10. The hypothesis is missing at the end of the introduction.

a. We have amended the end of the introduction to include a hypothesis

b. Revision (lines 78-84) - We performed a retrospective cohort analysis of patients admitted to member hospitals of a Louisiana healthcare system during various surges of COVID-19 to examine the differences in volume and outcome of patients admitted with COVID-19. To elucidate the effect of vaccination on outcomes, we compared patients admitted during the Delta and Omicron surges.

11. Line 80: Provide the age range for pediatric patients 

a. Revision (line 84): We also describe presenting symptoms of pediatric (ages 0 to 17) and adult (ages 18 and older) patients admitted to our tertiary referral center during the Omicron predominant surge. 

12. Line 84: List five hospitals that were involved. 

a. Revision (lines 89) - The Franciscan Missionaries of Our Lady Health System (FMOLHS) includes six hospitals that serve as regional referral centers in Louisiana and Mississippi; Our Lady of the Lake Baton Rouge, Our Lady of the Lake Ascension, Our Lady of Lourdes, Our Lady of Angels, St. Francis Medical Center and St. Dominic’s Hospital. The former five hospitals located throughout Louisiana share a common medical record and were included in the analysis.

13. Lines 89-92: How did you define the dates for each surge? need the references for each period

a. Author comment: We used our own health systems testing data which varies by geographic location and found the inflection point for each surge by percent positivity

14. Line 91: Be consistent with how you write the dates. 

a. Author comment – revisions made to December 15th 2021 (line 98)

15. Lines 96-97: Add references. Be specific with where the data is obtained. Is it from the Louisiana Department of Health as indicated by the asterisk? If so, why is it “assumed” if it is unbiased data obtained from the public institution?

a. Author comment – The data was never produced by the state or the CDC although the CDC was performing sequencing and communicating with state health departments during the first 3 surges. This state was in communication with our hospital’s infection prevention department to confirm that there was no new variant being sequenced during these times which was notable as there was evidence of new variants in other regions of the world during that time.

b. Data reported by the CDC was used for prevalence of delta and omicron during surges 4 and 5. We have cited this data as below. However, the Louisiana department of health via direct communication, not public reporting, assisted our research group in providing the percentages during the peaks of surge 4 and 5 for our state specifically.

c. Revisions (line 105) – At the peak of this surge, 98.13% of all Louisiana test isolates were identified as the Delta variant [CDC variant tracker]. In the winter of 2021, Omicron became the dominant strain in Louisiana, accounting for 98.99% of sequenced strains at the surge’s peak [CDC variant tracker]. 

i. CDC COVID Data Tracker: Variant Proportions ¬

16. Line 102: Include the IRB protocol number to show the study has been approved by IRB

a. Revision made (line 109) - This retrospective study was approved by the Louisiana State University Health Sciences Center – New Orleans Institutional Review Board (IRB #684)

17. Lines 106-107: “SARS106 CoV-2 testing was not required for all admissions but was recommended for any patient 107 presenting with symptoms consistent with COVID-19

a. Addition made to limitations (line 340) - Selection bias may be present given that COVID-19 testing was not performed on all admitted patients.

18. Line 119: needs to explain further what kind of vaccine the patients got. (i.e., mRNA, virus vector) Because the Jassen vaccine only requires one shot as opposed to others.

a. Author comment – Revisions were made to clarify vaccination categories and types of vaccines listed. We did not attempt to compare efficacy of one primary vaccine series to another.

b. Revision (Lines 132) - Unvaccinated patients had no record of vaccination or only received 1 dose of a 2 dose mRNA primary vaccine series. Patients overdue for booster completed a 1 dose virus vector or 2 dose mRNA primary vaccine series and were eligible for an additional immunization but had not yet received the additional dose.

19. Line 127: Revise the definition. There should be an updated version.

a. Author comment: This is the definition of sepsis we used while manually reviewing vital signs for chart abstractions. If the definition is changed, then the corresponding abstracted data will no longer be valid. 

20. Lines 128-129: How did you define the septic shock? I guess using EMR data would be hard to code. Using EMR data can lead the sepsis data to be considered a septic shock because of the fluid support. Didn’t you use the ICD code for data analysis? If so, the definition of the septic shock should be changed

a. Author comment: ICD codes were not used to define septic shock, vital sign parameters for blood pressure and MAR for vasopressor use was manually reviewed for each patient in this subset. 

21. Explain how you calculate the sample size. A flow diagram with numbers of invited, enrolled, and excluded subjects would be helpful to the readers.

a. Author comment: A sample size was not calculated. Any patient with a positive COVID test from an FMOLHS ED or hospital was included. There were no invitations, enrollments, or exclusions. 

22. Add the phrase for statistical significance of p-value (i.e., a p-value lower than 0.05 was adopted as the level of significance). 

a. Revision (line 121) - A p-value less 0.05 was adopted as the level of significance.

23. The authors have to cite the paper of statistical method guideline (i.e., DOI: https://doi.org/10.54724/lc.2022.e3)

a. Added citations for one way ANOVA and Chi-Squared Test [12, 13]

24. Lines 153,156,185: overused the word “significantly” -Katie

a. Revisions made to lines 170 and 173 omitting the word significantly

25. Lines 175-185: Use a consistent format of indicating p-values- line 175 p. 0.005 line 177 p <.0005 p<.0005 line 182 p .03 line 183 p. 0.29)

a. Revisions made so that < or = used for each value and similar decimal point location

26. Table 2: The left side is left-aligned, but the right side of the table is center-aligned. It would be better to match the text alignment 

a. Revisions made – all text left-aligned

27. Table 2, Line 146: Use CCI (Charlson comorbidity index) for the column ‘patients with comorbidities vaccination status’ & describe it in the method part. Also, describe the vaccination type (mRNA, virus)

a. Author comment – The Charlson comorbidity index was not used to measure comorbidities. The model for which we listed comorbidities was structed similarly to the cited study from South Africa (https://doi.org/10.1001/jama.2021.24868) in which comorbidities were pulled from certain ICD10 codes – if a patient had any of the identified ICD10 codes then a designation of cormobidity was made. 

b. Revision (line 213) – vaccine types specified more clearly 

c. *Below is a table of the ICD10 codes used to determine comorbidity, please let us know if this table should be included in the manuscript or as a supplement.

28. Line 228: add a comma after the introduction 

a. Revision made (line 251) - Following this variant’s introduction, hospitals

29. Line 230: “the” lowest

a. Revision made (line 253) - Although the Omicron surge resulted in the lowest admission

30. Lines 243 - 246: Not including the patients diagnosed and treated at home needs to be commented in the limitation 

a. Revision (line 342) - Additionally, patients diagnosed and treated at home were not included in this analysis.

31. Line 248: write out what LPM stands for before using it

a. Revision (line 273) - Changes in hospital practice also influenced certain clinical outcomes. For example, due to resource limitation during S4, the hospital policy allowed for up to 30 liters per minute of high-flow oxygen

32. Line 254: identify what “primary respiratory symptoms” are

a. Author comment: These were defined by ICD10 codes listed under admitted diagnosis and included codes for pneumonia, ARDS, pleural conditions, etc. 

b. Revision (line 280) - The percent of patients admitted with primary respiratory condition per ICD10 code, decreased as the pandemic progressed, and this decrease was most pronounced in S5 compared to previous surges. 

c. Description added to table 1: 

d. *Please see table below for list of ICD10 codes used from comorbidities and respiratory symptoms – would you like us to include this in the manuscript or as a supplement?

33. Line 266: Add the reference to support with quantitative evidence

a. Author comment: The quantitative evidence is listed both in Table 2 and in the results section “adults in the delta vs. omicron surge comparison by vaccination status.” 

b. Revision made (Lines 291) to reference table 2 - Vaccination was clearly protective for overall risk of hospitalization in the Delta and Omicron surge; however, vaccination did not appear to be associated with the same degree of protection against severe illness once admitted (defined as ICU admission, mechanical ventilation, disposition, or death) during S5 compared to S4 (Table 2).

34. Line 269: Change Thirty-one percent to 31% 

a. Revision made (line 296) -31% of patients admitted during the Omicron

35. Lines 271-3: this sentence contradicts your argument above in regards to COVID variation. What is the purpose of this sentence?

a. Line 298: The variant type may have played a part in outcome, but the surge-to-surge comparison also reveals that overall vaccination status is significantly improved during the Omicron surge and may also have played a role in overall outcomes. 

b. Author comment: It was not our purpose to make a statement that the patient outcomes observed are due to viral variant effect alone. Moreover, this sentence is the more conclusive of our impression as to what factors played a role in the outcomes that was seen amongst variants/surges e.g. variant characteristics, vaccination status, therapeutics, increased medical knowledge, etc.

c. Revision (line 298) - Although we are unable to determine if variant type played a part in outcome, our surge-to-surge comparison reveals that overall vaccination rate significantly increased during the Omicron surge and may have played a role in overall outcomes.

d. Revision/addition (line 301) - Additional factors that changed as the pandemic progressed such as improved medical knowledge, standardized practices, community awareness as well as advancing therapeutics likely also played a role in outcomes. 

36. Line 273: vaccination rate not vaccination status; increased not improved 

a. Revision made (line 299) - overall vaccination rate significantly increased

37. Line 281: except mortality during S5? line 279?

a. Lines 303-309: Despite the large age gap, mortality was similar between the younger unvaccinated patients and the more elderly, vaccinated population during Delta supporting the protective effect of the vaccine. A younger age was protective in all outcomes except mortality during S5. During the Omicron surge, vaccination appeared to trend towards more protection than age as the vaccinated group was only 3 years older than the unvaccinated group, and vaccinated patients were more likely to be discharged to home. However, this difference did not reach statistical significance. 

b. Author – We deleted the following sentence as it causes confusion (line 311): A younger age was protective in all outcomes except mortality during S5.

c. The following lines were added to better explain age findings in S5 (lines 315) - Additionally, during S5, the vaccinated patients requiring ICU admission were only 2 years older than vaccinated patient and were 8% less likely to require ICU admission, albeit this percentage difference did not reach statistical significance. And while also not statistically different, the vaccinated group was only 6 years older but 5% less likely to die than the unvaccinated group during S5. 

38. Line 280,281: Be consistent. Use either S5 or Omicron. Mixed terminology may confuse the reader.

a. Revisions made to use S4 or S5 instead of delta or omicron (lines 310-312)

39. Line 287: Change parentheses to bracket and put period after the bracket

a. Revision made (line 320) - Previous pediatric data collected from five pediatric hospitals during the Delta surge showed 29.5% of pediatric patients with COVID-19 required an ICU admission [9].

40. Line 320: Observation? It’s rather a recommendation/suggestion

a. Revision made (line 358) -Subsequent surges with potentially new variants are an expected reality, leaving us with two recommendations

---

## [Decision Letter · Decision Letter 1]

29 Aug 2022

PONE-D-22-13503R1Clinical Characteristics and Outcomes of SARS-Cov-2 B.1.1.529 Infections in Hospitalized Patients and Multi-Surge Comparison in LouisianaPLOS ONE

Dear Dr. Rivere,

Thank you for submitting your manuscript to PLOS ONE. After careful consideration, we feel that it has merit but does not fully meet PLOS ONE’s publication criteria as it currently stands. Therefore, we invite you to submit a revised version of the manuscript that addresses the points raised during the review process.

We look forward to receiving your revised manuscript.

Kind regards,

Dong Keon Yon, MD, FACAAI

Academic Editor

PLOS ONE

Journal Requirements:

Additional Editor Comments:

Please address comments of the reviewer.

Reviewers' comments:

Reviewer's Responses to Questions

**Comments to the Author**

1. If the authors have adequately addressed your comments raised in a previous round of review and you feel that this manuscript is now acceptable for publication, you may indicate that here to bypass the “Comments to the Author” section, enter your conflict of interest statement in the “Confidential to Editor” section, and submit your "Accept" recommendation.

Reviewer #1: All comments have been addressed

Reviewer #2: All comments have been addressed

2. Is the manuscript technically sound, and do the data support the conclusions?

Reviewer #1: Yes

Reviewer #2: Yes

3. Has the statistical analysis been performed appropriately and rigorously? 

Reviewer #1: Yes

Reviewer #2: Yes

4. Have the authors made all data underlying the findings in their manuscript fully available?

Reviewer #1: Yes

Reviewer #2: Yes

5. Is the manuscript presented in an intelligible fashion and written in standard English?

Reviewer #1: Yes

Reviewer #2: Yes

6. Review Comments to the Author

Reviewer #1: Dear author

I think the authors have adequately addressed all comments raised in a previous round of review.

Regards

Reviewer #2: The comments have been addressed, however, I encouragely suggest to be more conditional in the conclusions in the abstract, since there are associations, but not way to prove causality, considering issues with the changes in vaccination rates and determination of virus variants. In the same way, to specify that all the cases admitted to the hospital were considered, to clarify that there is not a sample size calculation (and the implications of that to make statistical inference).

7. PLOS authors have the option to publish the peer review history of their article (what does this mean?). If published, this will include your full peer review and any attached files.

Reviewer #1: No

Reviewer #2: No

---

## [Author Response · Author response to Decision Letter 1]

5 Oct 2022

Author Response to Reviewers

Please note that the lines referenced in the author comments refer to that of the revised manuscript with tracked changes and no longer correlate with the lines referenced by the reviewer comments as line numbers changed due to edits.

Responses to Reviewer #2 Comment:

Reviewer #2: The comments have been addressed, however, I encouragely suggest to be more conditional in the conclusions in the abstract, since there are associations, but not way to prove causality, considering issues with the changes in vaccination rates and determination of virus variants. In the same way, to specify that all the cases admitted to the hospital were considered, to clarify that there is not a sample size calculation (and the implications of that to make statistical inference).

Author Response:

1. Abstract conclusion has been edited to make conclusions more conditional/associations and less causal.

a. Revision (Lines 44-48): As the COVID-19 pandemic progressed, a younger and less vaccinated population was associated with higher risk for severe disease, fewer patients required ICU admission and overall mortality decreased. Vaccinations seemed to be protective for overall risk of hospitalization but once admitted did not seem to confer protection against severe illness during the omicron surge. Age also contributed to patient outcomes.

2. Methods section has been edited to clarify that sample size calculation was not performed.

a. Revision (Lines 94-96): All COVID-19 admissions and emergency department (ED) visits during the time periods corresponding to the individual surges of COVID-19 activity within the state of Louisiana were reviewed and considered; no sample size calculation was performed.

---

## [Editor Report · Decision Letter 2]

10 Oct 2022

Clinical Characteristics and Outcomes of SARS-Cov-2 B.1.1.529 Infections in Hospitalized Patients and Multi-Surge Comparison in Louisiana

PONE-D-22-13503R2

Dear Dr. Rivere,

We’re pleased to inform you that your manuscript has been judged scientifically suitable for publication and will be formally accepted for publication once it meets all outstanding technical requirements.

Kind regards,

Dong Keon Yon, MD, FACAAI

Academic Editor

PLOS ONE

Additional Editor Comments (optional):

This is an excellent paper.
---

## [Editor Report · Acceptance letter]

13 Oct 2022

PONE-D-22-13503R2 

Clinical Characteristics and Outcomes of SARS-Cov-2 B.1.1.529 Infections in Hospitalized Patients and Multi-Surge Comparison in Louisiana 

Dear Dr. Rivere:

I'm pleased to inform you that your manuscript has been deemed suitable for publication in PLOS ONE. Congratulations! Your manuscript is now with our production department. 

Kind regards, 

on behalf of

Dr. Dong Keon Yon 

Academic Editor

PLOS ONE